# Advanced Methods for Studying Structure and Interactions of Macrolide Antibiotics

**DOI:** 10.3390/ijms21207799

**Published:** 2020-10-21

**Authors:** Tomislav Jednačak, Ivana Mikulandra, Predrag Novak

**Affiliations:** Department of Chemistry, Faculty of Science, University of Zagreb, Horvatovac 102a, HR-10000 Zagreb, Croatia; ivana.mikulandra@chem.pmf.hr

**Keywords:** structure characterization, macrolide antibiotics, macrolide interactions, biomolecular targets, X-ray crystallography, cryo-electron microscopy, NMR spectroscopy, biochemical and fluorescence methods, molecular dynamics simulations

## Abstract

Macrolide antibiotics are macrocyclic compounds that are clinically used and prescribed for the treatment of upper and lower respiratory tract infections. They inhibit the synthesis of bacterial proteins by reversible binding to the 23S rRNA at or near the peptidyl transferase center. However, their excellent antibacterial profile was largely compromised by the emergence of bacterial resistance. Today, fighting resistance to antibiotics is one of the greatest challenges in medicinal chemistry. Considering various physicochemical properties of macrolides, understanding their structure and interactions with macromolecular targets is crucial for the design of new antibiotics efficient against resistant pathogens. The solid-state structures of some macrolide-ribosome complexes have recently been solved, throwing new light on the macrolide binding mechanisms. On the other hand, a combination of NMR spectroscopy and molecular modeling calculations can be applied to study free and bound conformations in solution. In this article, a description of advanced physicochemical methods for elucidating the structure and interactions of macrolide antibiotics in solid state and solution will be provided, and their principal advantages and drawbacks will be discussed.

## 1. Introduction

Owing to their high efficacy and safety, macrolide antibiotics have been in widespread clinical use for over 50 years. They belong to the largest class of antibiotics and are especially indicated for the treatment of upper and lower respiratory tract infections [1,2]. Macrolides target the bacterial (70S) ribosome, which consists of two subunits: small (30S), where the genetic code is translated into the sequence of amino acids, and large (50S), which is responsible for the peptide bond formation [3,4]. The reversible binding of macrolides to the 23S rRNA of the 50S subunit, at or near the peptidyl transferase center (PTC) blocks the exiting tunnel for newly synthesized peptides and thus inhibits the synthesis of bacterial proteins. The chemical structure of clinically relevant macrolides is characterized by a macrolactone ring, usually containing 14–16 atoms, substituted by polar and non-polar groups and linked to one or more sugars via glycosidic bonds. Smaller (14- and 15-membered) macrolides are primarily used for antibacterial therapy on humans, while those with a 16-membered ring have found applications in veterinary medicine [5]. Historically, this class of antibiotics can be divided into four generations, each having distinct structural features (Figure 1). The most widely used macrolide antibiotic is a first-generation erythromycin A, which was isolated in 1952 from *Saccharopolyspora erythraea* [6]. Since the discovery of erythromycin A, intense efforts have been made to prepare its semi-synthetic analogues or derivatives with enhanced bioactivity and physicochemical properties. As the result of these efforts, macrolides of the second generation, such as azithromycin, have been discovered, showing improved gastrointestinal tolerability, stability, and pharmacokinetics, but exhibiting cross-resistance with erythromycin A [7]. Telithromycin, a third-generation macrolide with an alkyl–aryl side chain at the macrolactone ring and a keto group instead of cladinose, showed activity against some resistant strains. However, telithromycin was withdrawn from the market after reports on severe side effects, including cardiovascular and liver toxicity, accommodation difficulties, blurred vision, nausea, and diarrhea, which is attributed to its pyridine moiety [8]. Having this in mind, a fourth-generation macrolide, solithromycin has been evaluated. Solithromycin has a similar structure to telitromycin, with major modifications in the side chain, where imidazole and pyridine moieties are replaced by 1,2,3-triazole and aniline rings, respectively. Furthermore, its macrolactone ring is substituted by a fluorine atom at position 2, which significantly improved activity against multidrug-resistant pathogens. Although this fluoroketolide was in late stage development, it did not obtain FDA approval due to suspected toxicity [9].

Overcoming resistance to antibiotics is one of the greatest challenges in modern medicinal chemistry. Since the number of newly discovered macrolides is decreasing, a permanent strategy, which comprehends basic principles of interactions with bacterial ribosome, is required to ensure their efficacy. Structures of some macrolide-ribosome complexes have recently been solved by X-ray crystallography and cryogenic electron microscopy (cryo-EM), throwing new light on the binding mechanisms of macrolides to the ribosome and providing a good basis for the design of new ligands and inhibitors [10,11,12,13,14,15]. This is a very important research field to the worldwide scientific community since the Nobel Prize for 2009 in Chemistry was awarded to groups studying the ribosome and its complexes with antibiotics.

However, when analyzing macrolide–ribosome complexes, one should take into account the discrepancies between the structures obtained for different species. It is also important to keep in mind that the structural features of the complex may not be exactly the same in solution and in the solid state. Hence, steps taken in the process of drug design should also include determination of the solution-state structure and dynamics of free and bound macrolides, their receptors, and their complexes. Free and bound macrolide conformations can be studied by applying an approach that combines NMR and molecular modeling calculations [16,17,18,19,20,21]. Since macrolide antibiotics may have different physicochemical properties and bioactivity, a detailed description of their structure and interactions with macromolecular receptors is crucial for assessing the overall biological profile. It can provide an insight into the macrolide binding mechanisms and facilitate the elucidation of biological effects exerted by this class of antibiotics on the human body. Key steps in this process involve the identification of functional groups and structural elements responsible for biological activity.

Structure and interactions of macrolide antibiotics can be studied by different experimental and theoretical methods. Basic principles of the advanced experimental tools for macrolide studies are displayed in Figure 2. Each method is based on different physical phenomena, providing a valuable, complementary data for the structural characterization of ligands, their receptors, and complexes.

In X-ray crystallography, the crystalline atoms cause a high-intensity X-ray beam to diffract into specific directions, forming a diffraction pattern (Figure 2a). By measuring the angles and intensities in the diffraction pattern, a three-dimensional electron density map can be produced. From this map, the mean atom positions, bond lengths, and other information can be determined [22]. On the other hand, cryo-EM allows a structural analysis of complex biological matter in a native environment using a coherent electron beam generated by field emission guns (Figure 2b). The beam is focused on a frozen sample solution embedded in a thin, vitreous layer of ice at the temperature around −180 °C. After interacting with the sample, the transmitted beams are directed to an electron detector and transformed into images [23]. Another analytical approach is based on NMR spectroscopy, which is a powerful method for probing macrolide interactions, since it can provide a wealth of information at nearly physiological conditions [24]. As shown in Figure 2c, the solution or solid-state sample in the magnet absorbs radiofrequency (rf) radiation. The obtained signal is transferred via NMR probe, amplified, and registered at the console as free induction decay (FID). Fourier transformation (FT) of FID results with an interpretable NMR spectrum of the sample.

Apart from the aforementioned methods, many biochemical and biophysical assays have been employed for macrolide binding studies, such as fluorescence spectroscopy, chromatography, gel electrophoresis, footprinting, toeprinting, calorimetric, and kinetic analysis. Most of these techniques require time-consuming analyte derivatization steps, which may change drug activity or separation steps and might affect binding equilibrium. Furthermore, single-molecule and bulk experiments can yield significantly different structural, kinetic, and thermodynamic information on the complex. In order to bridge this gap, experimental techniques outlined above are usually complemented with molecular dynamics (MD) simulations, which have become an essential tool for the accurate description of the structure, dynamics, and configuration of the ligands and complexes at atomic resolution [25].

In this review, recent applications of the advanced experimental and computational tools for studying the structure and interactions of macrolide antibiotics are described, and their main advantages and drawbacks are discussed.

## 2. X-ray Structure Characterization of Macrolide-Ribosome Complexes

X-ray crystallography is a powerful tool to study structural and molecular interactions and provides valuable data at the atomic level. Ribosome crystallography resulted in relevant models of macrolide-ribosome complexes for a very limited number of bacterial species, e.g., those amenable to crystallization, such as halophilic archaeon *Haloarcula marismortui*, Gram-positive *Deinoccocus radiodurans*, Gram-negative *Escherichia coli*, and thermophilic bacteria *Thermus thermophilus*. However, owing to the fact that the ribosome structure is highly conserved, the obtained structures have been frequently used to mimic complexes of macrolides with pathogenic bacterial ribosomes. Macrolides share a common characteristic to inhibit bacterial protein synthesis by binding to the bacterial ribosome at or near the PTC region. The assembled polypeptide chain leaves the ribosome through the nascent peptide exit tunnel (NPET) connecting the PTC. Macrolide antibiotics bind in the NPET, blocking the translation process usually at the early stage of protein synthesis.

There are several macrolide compounds that successfully co-crystallized with the bacterial ribosome and their three-dimensional structural network is elucidated. Hence, complexes of erythromycin and telithromycin (14-membered macrolides) with the 50S ribosomal subunit of *D. radiodurans* were determined by Yonath et al. [10,26], while structures of azithromycin (15-membered macrolide), carbomycin, spiramycin, and tylosin (16-membered macrolides), bound to the 50S ribosomal subunit isolated from *H. marismortui* have been reported by the Steitz group [11,27]. Those discoveries lead to the Nobel Prize in Chemistry for 2009. However, discrepancies between crystal structures of azithromycin bound to the *H. marismortui* and *T. thermophilus* ribosomes [28] on one hand and those observed for the large ribosomal subunit of *D. radiodurans* on the other have been noticed, although their ribosomes are highly conserved. The differences were also observed in the orientations and conformations of the azithromycin and erithromycin lactone rings. Furthermore, a cladinose sugar in erythromycin was found to be in low-energy chair conformation both in the free state and in the complex with the *H. marismortui* and *T. thermophilus* ribosomes, but in the complex with the large ribosomal subunit of *D. radiodurans*, it adopted the less energetically favored boat conformation. It was later suggested that this might be due to the different crystallization procedures used to crystallize azithromycin and the *D. radiodurans* 50S ribosomal subunit and the quality and interpretation of electron density rather than species specificity [28]. Complexes of azithromycin with the 50S ribosomal subunits isolated from *T. thermophilus* and *H. marismortui* display a common set of contacts between the macrolactone ring and the hydrophobic surface of the ribosomal exit tunnel (U2611, A2058, and A2059). The structures of these antibiotics bound to the *H. marismortui* large subunit resemble those observed in the free state.

Previous studies demonstrated that 14- and 15-membered macrolides may exist in the two main conformations, the folded-out and the folded-in both in solution and solid state [16,17,18,19]. These conformations refer to the outward and inward folding of the macrolactone ring fragment C3–C5 (Figure 1).

The folded-in conformation is characterized by closer contacts between H3 and H11 atoms and between the CH_3_ group at position 6 and the H4 atom, while in the folded-out conformation, the H4 atom is closer to H11 and H5 to the methyl group at position 6. The main distances defining the folded-out conformation of both free and bound erythromycin and azithromycin are displayed in Table 1. It is clearly seen that distances C4–C11 are shorter than C3–C11 for unbound azithromycin and erythromycin, which was also found in the complexes with the *H. marismortui* and *T. thermophilus* ribosomes (Figure 3). It is a common characteristic for the folded-out conformation. Furthermore, much closer contacts C5–C6Me than C3–C8 are in accordance with this observation. NMR methods revealed that in solution, the predominant conformation was folded-out, but portions of folded-in could also be present depending on the solvent polarities [16,17,18,19]. It was found that 15-membered macrolides have more conformational flexibility than 14-membered ones. They could also adopt the 3-endo-folded-out conformation, which has characteristics of both conformational families. The presence of two forms of this structure even in the crystal reflects the conformational flexibility of these compounds. When bound to the *E.coli* ribosome, the macrolide conformations mostly resemble those in the free state. However, it is somewhat puzzling that for azithromycin complexed with the *H. marismortui* ribosome, structures with different C4–C11 and C3–C11 distances have been reported depending on the resolution (Table 1). In the structure obtained with higher resolution, C3–C11 was shorter than C4–C11, reflecting the presence of the folded-in or the 3-endo-folded-out conformer as found in solution [19]. The same was found for the complex with the *T. thermophilus* ribosome. This would imply a certain flexibility of the macrolactone ring of the 15-membered macrolides in the bound crystal structure as well, which probably contributes to their binding affinity and pharmacokinetics.

Hence, the structural models showed that the folded-out conformation is the predominant bound conformation [27,28]. When complexed with ribosome, the polar groups of the macrocyclic lactone ring are directed toward one side of the molecule, while the other side is mostly hydrophobic. The polar macrolide region is in close contact with the lumen of the ribosome exit tunnel, while the hydrophobic part is oriented toward the tunnel wall.

The first ribosome–macrolide structures for pathogenic bacteria were reported for the *E. coli* 50 S ribosomal subunit [12]. Binding modes of erythromycin and telithromycin showed significant variations compared to previously reported structures observed for non-pathogenic strains, which was in accordance with biochemical studies (Figure 4). Hence, crystallography provided important structural information on macrolide–ribosome complexes; however, the majority of them were observed for clinically non-relevant bacteria, and the question arises as to whether these could be taken to explain all the effects of macrolides on different pathogenic strains. May species specificity account for the observed differences? In addition, solid and solution-state structures may not be entirely the same, and this should be taken into account when designing novel macrolides with bioactivity.

Kannan and Mankin emphasized that most of the proposed structural models neglected the species-specific mode of action, which may produce significant differences in drug binding structures [31]. Examples were presented showing how the binding of macrolides and associated bacterial resistance can be affected by the species-specific properties of ribosome architecture. The variations in ribosome structure and differences in macrolide binding modes can accelerate efforts on designing more effective macrolide antibiotics.

Svetlov et al. [13] recently reported on high-resolution crystal structures of erythromycin and chloramphenicol bound to the *T. thermophilus* 70S ribosome allowing an unambiguous placement of the drugs in the electron density maps, which provided evidence for direct collision of the two drugs and rationalized the observed competition between them. This fact was not noticed in the previous study [26], owing to insufficient resolution. The binding of erythromycin was accomplished by H-bonding between bases A2058 and A2059 and the desosamine sugar of the drug. Superimposed structures of the two molecules show that steric clash occurs between the chlorine atoms of the chloramphenicol and dimethylamino group of desosamine sugar in erythromycin. It was also indicated that erythromycin induced a conformational change of A2062 in the *T. thermophilus* ribosome, which was not observed with *E. coli*, which might be due to species-specific binding.

Complexes of methymycin and pikromycin, 12- and 14-membered macrolide and ketolide, respectively, with the *T. thermophilus* 70S ribosome have recently been reported [32]. A combined use of structural, genetic, and biochemical data revealed the binding of both compounds in the NPET region, thus inhibiting the progression of a nascent polypeptide chain after a limited number of proteins were synthesized. Authors commented on the differences between previously reported binding of methymicin at the A site of the PTC in the 50S ribosomal subunit of *D. radiodurans* as a consequence of either a species-specific mode of action or misinterpretation of the crystallographic data.

To better understand the role macrolide antibiotic binding protein-1 (MABP-1) plays in conferring resistance to erythromycin in *M. tuberculosis*, Zhang et al. [33] determined the crystal structure of the complex. It was demonstrated that the complex was stabilized by hydrogen bond and hydrophobic interactions and that erythromycin was located in a shallow pocket formed by both subunits of the homodimer. In order to explore the macrolide specificity of MABP-1, authors investigated clarithromycin and azithromycin and demonstrated that the protein had a preference for 14-membered clarithromycin rather than 15-membered azithromycin macrolide. Based on the obtained results, they speculated that the protein could dissociate erythromycin from the ribosome as suggested previously [34], or alternatively, it may cooperate with efflux pumps that expel the antibiotic.

Fong et al. presented the structures of macrolide kinases conferring resistance to macrolides in complex with six different macrolide antibiotics [35]. The structures showed that an extended and flexible interdomain linker was responsible for the binding of those antibiotics. The authors stressed that a comparison of macrolide binding to kinases and the bacterial ribosome can be an avenue for the development of next-generation macrolide antibiotics.

X-ray crystallography was also used to determine the absolute configuration of several 12-, 14-, and 16-membered macrolide compounds isolated from fungi species [36,37,38]. Bastimolide A, a polyhydroxy macrolide with a 40-membered ring isolated from cyanobacterium *Okeania hirsuta* was structurally characterized by X-ray diffraction analysis, and its absolute conformation was determined [39].

## 3. Cryo-EM Imaging of Macrolide-Targeted Ribosomes

Cryo-EM enables high-resolution imaging of biological macromolecules and their complexes in a fully hydrated state, without causing interferences by crystalline packing. The recent development and application of direct electron detectors have made cryo-EM one of the most valuable techniques for the structural determination of dynamic macromolecular assemblies, such as protein complexes, viruses, or ribosomes and for the visualization of conformational changes upon ligand binding. Apart from its importance in structural and functional studies of biomolecular complexes, cryo-EM also plays an integrative role at the interface with X-ray crystallography, scanning electron microscopy (SEM), and fluorescence microscopy [40,41]. In modern drug discovery, cryo-EM has already found applications in target identification, validation, and characterization, which usually includes structural analysis of the receptor, ideally in the complex with a drug-like molecule. Owing to its current reproducibility, quality, and throughput, this approach can be used in fragment-based screening and structure-based drug design. Excellent discussions about the applications of cryo-EM in drug discovery can be found in a number of articles [42,43,44,45,46].

According to the procedure used for data collection, cryo-EM methods can be divided into two main groups: single-particle analysis (SPA) and cryogenic electron tomography (cryo-ET) [47]. SPA is based on the alignment and averaging of many thousands of images of isolated macromolecules and complexes in vitro, followed by the reconstruction of three-dimensional maps at near-atomic resolution. Structural information in SPA is collected from multiple untilted copies of the same structure. On the other hand, cryo-ET imaging is carried out on a single area of the sample, but from different angles, allowing the structural elucidation of heterogeneous and complex entities, e.g., prokaryotic cells. Despite having different experimental procedures, SPA and cryo-ET share similar issues encountered during sample preparation, including aggregation, denaturation and the inadequate density or thickness of a biological material [47]. An additional limiting factor is the time-consuming, expensive, and labor-intensive sample screening on electron microscopes. Practical solutions for this limitation can be obtained by applying process optimization and automation technologies [48].

SPA has recently emerged as a powerful tool for studying ribosomes and their interactions with inhibitors, substrates, and regulatory molecules. High-resolution cryo-EM maps have been reported for bacterial ribosomes [49,50,51], providing structural and functional insights into the species-specific mechanisms of protein translation, resistance, and ligand binding. A similar approach has been applied to structurally characterize macrolide–ribosome complexes, visualize the binding site, and pinpoint conformational changes of the interacting residues [14,15]. Specific interactions of dirithromycin with the *E. coli* ribosome observed by SPA provided a rational explanation for enhanced activity of the drug in comparison with its parent compound, erythromycin [14]. Furthermore, Halfon et al. employed the same method to investigate the effect of Arg88–Ala89 deletion in the uL22 β-hairpin loop of the ribosomal protein from *S. aureus* on erythromycin activity [15]. It was shown that the deletion induced structural changes in the ribosome, conferring resistance to erythromycin, but it had little effect on macrolide binding. In the cryo-EM structure of erythromycin bound to the large ribosomal subunit of *S. aureus* (PDB ID: 6S0Z, resolution 2.30 Å) [15], the C3–C11 (5.35 Å) and C4–C11 (5.30 Å) distances were in accordance with the crystallographic data obtained for folded-out conformations [11,28]. On the other hand, in the complex of erythromycin with the whole *S. aureus* ribosome (PDB ID: 6S0X, resolution 2.42 Å) [15], the distance between C4 and C11 atoms was found to be 5.55 Å, while the C3–C11 distance (5.37 Å) was similar to that observed in the complex with the 50S subunit alone (5.35 Å). As elaborated previously, these differences indicate that the conformational flexibility of erythromycin is affected by interaction specificity.

Another research field where cryo-EM has been utilized is the imaging of stalled ribosome complexes with erythromycin-dependent nascent peptides [52,53,54]. In the presence of erythromycin, the translation of nascent erythromycin resistance methyltransferase B (ErmBL) and C (ErmCL) leader peptides encoded by erythromycin resistance methyltransferase (*erm*) genes is stalled at the specific positions in the open reading frame. Conformational changes of the mRNA triggered by the stalling release the ribosomal binding site for *erm* and activate expression of the resistance gene [55]. The structural basis for ErmBL-mediated translational arrest was studied by Arenz et al., who combined cryo-EM and MD simulations to explain the stalling mechanism [52,53]. They identified a complex hydrogen bond network between the C-terminal region of ErmBL and nucleotides of the 23S rRNA, reflecting the importance of these interactions for translational arrest. The nascent peptide adopted a previously unseen conformation with the Asp10 side chain in a rotated position as the result of erythromycin binding. Inhibition of the peptide bond formation was explained by allosteric interactions, which prevented the transfer of aminoacyl-tRNA (aa-tRNA) to the peptidyl site and perturbed the orientation of peptidyl-tRNA (p-tRNA) by restricting the path of ErmBL. Similar conformational rearrangements were observed in the cryo-EM structure of the ErmCL-stalled ribosome reported by the same group [54]. In the presence of erythromycin, ErmCL interacted directly with the drug and prevented stable binding of the aa-tRNA at the PTC, leading to translational inhibition.

An alternative stalling mechanism is based on interactions of the tryptophanase leader peptide (TnaC) with the exiting tunnel in the presence of free tryptophan. The TnaC-mediated translational inhibition was studied by Seidelt et al., who carried out SPA of the *E. coli* ribosome-TnaC complex stalled by the tryptophan addition [56]. It was demonstrated that the stalling induced changes in the orientation of conserved residues A2602 and U2585 within the PTC, which prevented the binding of release factors required for the translation progress.

Several research groups applied cryo-EM to characterize different forms of tRNA-bound ribosomes. The incorporation of aa-tRNA into the *E. coli* ribosome was investigated by Valle et al. [57]. Cryo-EM maps revealed that the flexibility of aa-tRNA facilitated its accommodation into the PTC and enabled the initial codon recognition. On the other hand, Morgan et al. combined cryo-EM with three-dimensional variability analysis to characterize tRNA-bound states of the *A. baumannii* ribosome and identify interparticle motions between the 30S and 50S subunits [58]. Structural data obtained by this approach clarified the mechanism of protein translation in this pathogenic strain and can be further exploited for the development of ribosome-targeted therapeutics.

Considerable roles in the translation inhibition and bacterial resistance play ribosomal silencing factors (RSF) and ATP-binding cassette (ABC) proteins, respectively. Li et al. solved the cryo-EM structure of the *M. tuberculosis* 50S subunit bound to RSF and the crystal structure of a free protein [59]. Chromatographic and crystallographic data pointed toward the presence of RSF dimer in both solution and solid state. Surprisingly, the same protein existed as a monomer in the cryo-EM structure of the RSF-bound large subunit, indicating that the RSF dissociation is required for effective translation inhibition. In some cases, ABC proteins prevent the inhibition of translation by dislodging a macrolide from its target site [34]. Su et al. reported the cryo-EM structure of the ABC-F type macrolide-streptogramin B resistance protein (MsrE) in complex with the *T. thermophilus* ribosome and proposed the mechanism of ribosome protection [60]. It was found that the MsrE protein formed a needle-like arrangement with two transporter ABC domains carrying conserved nucleotide-binding sites and a domain linker composed of two crossed helices connected by an extended loop. If MsrE binds to the ribosome, the extended loop mediates conformational changes in the PTC, p-tRNA, and exiting tunnel, leading to the release of a macrolide.

## 4. Probing Macrolide Interactions by NMR Spectroscopy

NMR spectroscopy is an indispensable method for the three-dimensional structure elucidation of drug-like molecules and probing their interactions with biological receptors in solution. There are numerous one- and two- dimensional NMR techniques that give a plethora of information on biological complexes. By using NMR experiments, it is possible to elucidate free and bound ligand conformations, ligand binding sites, and the local mobility of the complex, ligand, and receptor. The evaluation of binding constants can also be achieved by different NMR techniques.

According to the parameters used for studying ligand–receptor interactions, NMR techniques can be divided into titration, diffusion, relaxation, and nuclear Overhauser effect (NOE)-based experiments (Figure 5).

With NMR titration experiments, one can detect ligand binding, which is manifested by line broadening and differences in the ligand chemical shifts [21]. Further information on ligand binding and estimation of the binding strength provide NMR experiments based on translational diffusion phenomena. If the complex is formed, motional properties of the ligand become similar to those of the macromolecular receptor. Diffusion coefficients of free and bound ligands can be measured by diffusion-ordered NMR spectroscopy (DOSY). Another NMR approach for studying macrolide interactions involves relaxation measurements. Ligand binding can be detected by changes in longitudinal (T_1_) and transversal (T_2_) relaxation times or paramagnetic relaxation enhancements (PREs). Solvent PREs are especially suitable for assessing the orientation, conformation, binding modes, and immersion depth of the ligand into the receptor. The experiment is performed with a water-soluble and inert paramagnetic agent, which generates a “paramagnetic solvent” and changes magnetic properties of the solution. Relaxation enhancements of the ligand nuclei depend on their distance from the “paramagnetic solvent”, i.e., on the immersion depth at the target. In a fast exchange regime between the free and bound states, solvent PREs can be transferred to the free ligand and detected [21]. The most widely used NMR methods based on NOE polarization transfer are saturation transfer difference (STD) and transferred nuclear Overhauser effect spectroscopy (trNOESY). STD is applied to identify the binding epitopes of ligands [61], while trNOESY allows the precise determination of free and bound ligand conformations. These techniques can also be useful for ligand design and for screening their biological activity.

As already mentioned, macrolides exert their biological activity by binding to the bacterial ribosome. According to the recent studies [62], binding mechanisms of macrolides to ribosomes depend on the structure of the macrolide and the bacterial strain studied. In order to completely understand the mechanism of action of macrolides, it is important to determine their conformation in solution and to investigate interactions with the ribosome. Over the last few decades, a lot of research has been directed toward the analysis of ribosome–macrolide complexes. Approaches that combine NMR and molecular modeling methods have been used extensively to study the conformational behavior of free and bound 14- and 15-membered macrolide antibiotics [16,17,18,19,20,21,62].

Everett et al. characterized the conformation of the first-generation macrolide, erythromycin A, by NMR spectroscopy [63]. Two conformational states were detected: folded-out and folded-in. Further investigations based on the analysis of NMR coupling constants and NOEs confirmed those two conformations of 14-membered macrolides in solution [2]. However, 15-membered macrolides could also adopt the 3-endo folded-out conformation, which possesses structural features of both folded-in and folded-out forms [19,64].

As previously mentioned, STD and trNOESY are two fast and robust methods that rely on the NOE effect and provide plenty of information on the molecular structure and interactions of macrolides with bacterial ribosomes. By monitoring NOE signals in trNOESY spectra, it is possible to estimate the three-dimensional macrolide conformation in the bound state (Figure 6). NOE signals are either positive or equal to zero for small molecules (*M* ≈ 1000 Da) because of their short correlation times. NOESY spectra of macrolides could display either positive (Figure 6a) or negative (Figure 6b) NOE cross peaks, depending on the solvent viscosity. On the other hand, when bound to the ribosome, macrolides exhibit longer correlation times than in the free state. Accordingly, they adopt the same NOE behavior as the receptor and show strong and negative NOEs [65].

NOESY experiments in combination with X-ray and molecular modeling have demonstrated that macrolides in a ribosome-bound state adopt similar conformation as in a solution free state [65]. In cases of zero NOEs, an alternative technique called rotating frame nuclear Overhauser effect spectroscopy (ROESY) is usually applied. However, with this technique, caution should be exercised owing to serious artifacts such as TOCSY–ROESY and ROESY–TOCSY correlations.

Another NOE-based technique for the characterization of ligand–receptor interactions is STD NMR spectroscopy [61]. This powerful technique is capable of identifying binding epitopes of a ligand when bound to its receptor. It is based on the selective saturation of receptor resonances and subsequent saturation transfer to the bound ligand by a spin-diffusion mechanism. The degree of saturation of the individual ligand protons depends on their spatial distance from the receptor surface and binding kinetics. STD takes the difference of two experiments: off-resonance, with the irradiation frequency set far from any signal and on-resonance, where receptor resonances are selectively saturated. A difference spectrum, showing only resonances that experience saturation upon ligand binding, is obtained by subtracting the on-resonance spectrum from the off-resonance one, as shown in Figure 7 for 4″-aminopropyl azithromycin, which is a precursor for obtaining the novel class of bioactive azithromycin conjugates, the macrozones [66].

Then, STD enhancements are calculated using Equation (1):(1)$$A_{STD}=\frac{I_{o}-I_{sat}}{I_{o}}=\frac{I_{STD}}{I_{o}}$$
where *A_STD_* is the STD amplification factor, and *I_o_* and *I_sat_* are peak intensities in off-resonance and on-resonance spectra.

The largest saturation transfer for 4″-aminopropyl azithromycin bound to the ribosome was detected for several methyl groups at positions 14, 8, and 4 and protons in desosamine (position 3′), cladinose (position 4″), and methylene protons (position 4″c) of the aminopropyl group (Figure 7) being consistent with previously reported results [19]. Hence, STD experiments revealed three common structural moieties of the 15-membered macrolides in close contact with the ribosome: cladinose, desosamine, and methyl group at the position 14 [19]. Regardless of the fact that the binding of other studied macrolides slightly differs depending on their structural characteristics, it was shown that the desosamine sugar unit is a common structural part that plays a significant role in macrolide binding to the ribosome [17,62]. Binding epitopes as determined by NMR were in accordance with those obtained by crystallography [11,12,19].

A fast exchange process between the bound and free state is a prerequisite for using these two NOE-based techniques. This usually corresponds to weak to moderate ligand binding. It has been shown that the binding of macrolide to the ribosome is an allosteric two-step process. The first step involves weak macrolide binding in fast exchange, i.e., the recognition of the target site on the ribosome. The second step involves strong binding to the active site of the ribosome responsible for macrolide activity [64,65].

To understand the overall biological profile of macrolides in the process of drug design, it is important to investigate interactions of macrolides with other biological targets, such as bacterial membranes, bile acids, and some relevant proteins [17,18,19,20,21]. This approach has been employed in drug development to optimize the pharmacokinetic profile of bioactive compounds by enhancing their solubility, bioavailability, distribution, and excretion. Therefore, the physicochemical properties of a potential drug candidate can be improved by accurate determination of the binding sites and their modes of action.

An interesting study of macrolide–membrane interations has been reported by Kosol et al., who investigated macrolide binding to membrane mimetics, such as sodium dodecyl sulfate (SDS) and dodecylphosphocholine (DPC) by several NMR techniques [20]. They used self-diffusion NMR experiments and solvent PRE to obtain data on the binding, orientation, and location of macrolide antibiotics in the membrane mimetics. The authors determined the interaction strength of several macrolides (active and inactive) with membrane mimetics by measuring self-diffusion coefficients before and after the addition of SDS and DPC micelles. A significant decrease in self-diffusion coefficients pointed toward macrolide binding to the membrane mimetics. By monitoring longitudinal relaxation in the presence of various concentrations of inert paramagnetic agent, they discovered that positively charged amino groups on desosamine sugar and the macrolide ring are the closest to the micelle surface. Authors also investigated the cause of a cellular disorder called phospholipidosis by using one-dimensional NMR spectroscopy. It is known that macrolides play a significant role in this process by blocking the action of enzyme phospholipase A1 responsible for phospholipid degradation. The mechanism of phospholipidosis and interactions of phospholipids with enzymes and azithromycin were successfully described by monitoring line broadening in ^1^H NMR spectra. They compared the ^1^H NMR spectra of phospholipids incubated with phospholipase A1 in the absence and presence of azithromycin. The results indicated that macrolide binding to lipids protected them from being degraded by phospholipase A1.

Owing to the fact that most macrolides are excreted by bile, Glanzer et al. studied interactions between different macrolides and bile acids by titration, diffusion, and relaxation NMR experiments [21]. Significant line broadening and shifts of proton resonances observed during NMR titrations of macrolides pointed toward their binding to the bile acid micelles. Diffusion experiments provided information on the strength of macrolide interactions with the bile acid micelles by the mole fraction partition coefficient *K_p_*, according to Equation (2):(2)$$K_{P}=\frac{D_{e}-D_{f}}{D_{mic}-D_{e}}=\frac{A_{b}}{A_{f}}$$
where *D_f_*, *D_e_*, and *D_mic_* are the diffusion coefficients of free macrolides, macrolides in bile salts, and bile-salt micelles. *K_p_* is defined as the ratio of bound and free ligand molecules. Furthermore, NMR experiments based on solvent PREs were applied to determine the binding modes and positioning of macrolides in the micelles. Similar solvent PREs for both bile acids and antibiotics across their entire structures indicated there were no preferred orientations of them in the complexes.

High bioavailability is one of the most important physicochemical properties of drug molecules. Since it greatly depends on the binding of drug molecules to plasma proteins, Novak et al. applied STD NMR spectroscopy to elucidate macrolide interactions with bovine serum albumin (BSA). They determined the binding epitopes of azithromycin, oleandomycin, and telithromycin. STD NMR spectra showed similar binding modes for azithromycin and oleandomycin. The binding modes for telithromycin were different, which was probably due to the different structural elements of these macrolides in close contact to BSA. The desosamine protons in azithromycin and oleandomycin intimately approached BSA, while in telithromycin, they were found to be more distant. The absence of a cladinose sugar unit and the presence of an alkyl-heteroaryl side chain in telithromycin caused differences in the binding mode [18].

Yuan et al. applied a combination of the attenuated total reflectance Fourier transform infrared spectroscopy, ^31^P NMR technique, and the molecular dynamics simulation to analyze the action mechanism of azalomycin F_5a_, which is a polyhydroxy macrolide that showed bioactivity against methicillin-resistant *Staphylococcus aureus* (MRSA) [67]. The specific interactions of azalomycin F_5a_ with polar phospholipid heads observed in ^31^P NMR spectra explained the previously reported increase of bacterial membrane permeability in the presence of the drug. Furthermore, the authors observed different phosphorus chemical shifts of model membranes in the presence and absence of azalomycin F_5a_.

Amphotericin B and natamycin are polyene macrolides used for fungal infection treatment. Their mode of action includes interactions with sterols in membranes. Ciesielski et al. demonstrated the application of solid-state magic angle spinning (MAS) ^13^C NMR experiments to study the effect of natamycin and amphotericin B on mixed lipid membranes containing dioleoylphosphatidylcholine (DOPC), cholesterol, and ergosterol [68]. It was found that amphotericin B penetrated into the lipid membrane in the absence of preferential interaction with either cholesterol or DOPC, leading to increased freedom of both DOPC and cholesterol. On the other hand, natamycin interacted with DOPC or cholesterol to a lesser extent. Amphotericin B/ergosterol pairing on opposite sterol faces was explained by a head to tail interaction model. Furthermore, Debouzy et al. utilized ^1^H NMR, ^31^P NMR, and electron spin resonance (ESR) techniques for studying the interactions between amphotericin B and phospholipids in the samples with various proportions of 1,2-dimyristoyl-sn-glycero-3-phosphocholine (DMPC) and 1,2-dimyristoyl-sn-glycero-3-phosphorylglycerol (DMPG) [69]. Their results indicated a good encapsulation of amphotericin B in sample containing equimolar amounts of DMPG and DMPC.

As stated above, NMR spectroscopy has frequently been used for identification and structure determination. Hence, in recent SAR (structure–activity relationship) profiling of irumamycin-type macrolides [70], azithromycin acylides [71] and a new dimeric macrolide cocosolide [72], one- and two-dimensional NMR techniques have been applied to determine their structure. As elaborated, the structure characterization of free and bound ligands can provide the details about ligand binding modes and other essential information for the design of new and more potent drugs. In order to fully determine the free and bound ligand conformations, it is important to establish the configuration of all stereogenic centers. Owing to the fact that macrolides are complex molecules with many chiral atoms and that an inversion of stereochemistry can affect their bioactivity and physicochemical properties, a group of researchers developed an NMR-based approach using a combination of NOE, *J*-couplings, and residual dipolar couplings (RDC) to elucidate a stereochemical configuration of archazolide A, a 24-membered macrolide [73].

Nakano et al. utilized NMR spectroscopy for conformational analysis of newly synthesized 14-membered macrolides [74]. Advanced NMR techniques were applied to confirm their structure, conformation, and calculate the diastereomer ratio. Similarly, Arsić et al. investigated the conformation of tylosin A, a 16-membered macrolide, in deuterated water. Since tylosin A exists in different forms depending on the solvent used, the authors employed molecular modeling and one- (^1^H and ^13^C) and two-dimensional (ROESY) NMR techniques, in order to identify tylosin A form and conformation in D_2_O [75]. Furthermore, a number of recent papers reported on the identification and structure determination of macrolide natural products by employing NMR spectroscopy. Examples are given in ref. [36,37,38,39,76,77,78,79,80,81,82,83,84,85].

Grgičević et al. have recently reported on the use of molecular modeling and NMR studies to design and characterize a novel hybrid class of bioactive azithromycin derivatives, the macrozones [66]. Interactions and binding properties of selected azithromycin–thiosemicarbazone conjugates to the ribosomal 23S rRNA were investigated by an in silico by docking study that led to the discovery of new potent macrolide compounds called macrozones.

## 5. Other Methods for Macrolide Binding Studies

Biochemical and physicochemical assays provide interesting alternatives for X-ray, cryo-EM, and NMR methods, as they do not require a complete structure characterization of ligands, receptors, or their complexes and offer a fast and efficient way to confirm binding, identify the binding site, and assess how the complex formation affects biomolecular structure and function. There are three major groups of commonly used assays for macrolide interaction studies.

The first group is based on footprinting experiments, which allow a quick and reliable assessment of macrolide binding sites [86]. In the footprinting assay, a modification agent is introduced in vitro into the mixture of ribosome and macrolide solutions (Figure 8a). After nucleotide modification, the rRNA is extracted and the positions of modification are identified by using primer extension and gel electrophoresis. Macrolide binding protects a part of the rRNA from the modification, revealing a footprint in the electropherogram of a modified complex. Several research papers published in the last decade consider the use of footprinting as the principal method for describing the interactions of macrolide antibiotics with bacterial ribosomes [87,88,89]. Xiong et al. characterized the binding site of 6,11-*O*-bridged macrolide derivatives at the *E. coli* ribosome by identifying their footprints after the treatment with dimethyl sulfate (DMS), which methylates the nitrogen atoms N1 in adenine and N3 in cytosine and with 1-cyclohexyl-3-(2-morpholinoethyl) carbodiimide metho-*p*-toluene sulfonate (CMCT), which modifies guanine and uracil residues at positions N1 and N3, respectively [87]. The results were comparable with competition binding experiments and indicated that the bridged derivatives and erythromycin interacted with the same or overlapping sites at the *E. coli* ribosome. Similarly, Poulsen et al. utilized the footprinting assay to study interactions of pleuromutilins and macrolides with the *E. coli* ribosome [88]. It was demonstrated that pleuromutilin derivatives and carbomycin competed for an overlapping binding site at the ribosome, while erythromycin targeted another region.

An interesting approach was used by Petropoulos et al., who combined footprinting and kinetic experiments to describe the entire course of azithromycin binding to the *E. coli* ribosome and to explore the effects of magnesium ions and polyamines on the reaction mechanism [89]. Kinetic data pointed toward a two-step binding process: azithromycin was first recognized by the ribosome machinery and placed in a low-affinity site within the upper part of the exiting tunnel, followed by the slow formation of a final complex. Furthermore, mutual exclusion between the low- and high-affinity sites at the *E. coli* ribosome prevented the simultaneous binding of two azithromycin molecules. That observation was in sharp contrast to the results obtained for *D. radiodurans*, where two cooperatively interacting sites for azithromycin existed in the ribosome. A high concentration of Mg^2+^ hindered drug binding to both sites, while polyamines blocked the shift of azithromycin to the high-affinity site and attenuated the complex formation.

In the toeprinting analysis, primer extension and gel electrophoresis are applied to measure the ribosome position on the mRNA [90]. If macrolide binding triggers translational inhibition, a specific band in the electropherogram (toeprint) is shifted from the toeprint of a ribosome at the stop codon, pointing toward the stalling position (Figure 8b). Toeprinting experiments can generally be used if the complex is stable enough to withstand displacement by reverse transcriptase during the primer extension. Representative examples are the stalled ribosome complexes with macrolides and nascent peptides mentioned in the previous section. Ribosome stalling at the ErmCL peptide was studied by the group of Vázquez-Laslop, who used a combination of toeprinting, selective amino acid labeling, and mutational analyses to assess the contribution of ErmCL, erythromycin, and the bacterial ribosome toward the formation of the stalled complex [55]. Toeprinting showed that stalling occurred at the Ile9 codon, which was further confirmed by selective amino acid labeling. Mutational analysis revealed that universally conserved residues of the 23S rRNA play crucial roles in the stalling mechanism. In their recently published paper [91], Koch et al. corroborated these results by toeprinting and atomic mutagenesis assays, highlighting A2062, A2053, and U2586 (*E. coli* nomenclature) as the key players in the ErmCL-mediated translational arrest. A similar methodology was applied by Sothiselvam et al., who explored how macrolides affect the binding of nascent peptides to different acceptor substrates at the ribosome [92]. They carried out mutational and toeprinting analysis and suggested that the binding of erythromycin in the exiting tunnel blocked the transfer of nascent tripeptides with an Arg or Lys as their penultimate residues to an Arg or Lys acceptor. The authors found that macrolide-induced translational arrest was affected by the length of amino acid side chains in the donor and acceptor substrates and the positive charge of critical residues.

An alternative approach to study macrolide interactions involves selective labeling of the ribosome or antibiotic with a suitable fluorophore (F) and subsequent fluorescence detection of the complex (Figure 8c). Johansson et al. have shown that fluorometric analysis can successfully be employed to monitor the elongation dynamics and the effect of erythromycin on ErmCL-mediated stalling [93]. For this purpose, they applied an assay based on Förster resonance energy transfer (FRET), which was induced by mutual rotation of the small and large subunits labeled with fluorescent donor (Cy3 or Cy3B) and acceptor (Cy5) dyes, respectively. Real-time tracking of the elongation stage facilitated explanation of the stalling mechanism. The authors emphasized that erythromycin binding triggered an abrupt ribosome stalling, in contrast to the gradual slowdown and heterogeneous stall positions observed in the macrolide-independent process. Complementary information on the binding dynamics can be obtained by fluorescent labeling of the drug. Tereshchenkov et al. prepared a series of fluorescent macrolide derivatives and calculated the dissociation constants for their ribosomal complexes from the fluorescence polarization data [94]. It was found that certain tylosin derivatives with fluorescent groups could be used as probes to characterize the binding of new macrolides and other drugs to the exiting tunnel. However, these results should be interpreted with caution, having in mind that the binding affinity and inhibitory activity of macrolides may differ from their fluorescent derivatives.

## 6. Computational Simulations of Macrolide Interactions

In the last few decades, computational simulations have become an integral part of drug design strategies and allowed a deeper understanding of molecular mechanisms and dynamics involved in ligand-receptor interactions. MD simulations of macrolides, their biological targets, and complexes are usually combined with experimental techniques to provide valuable information on the molecular structure, conformation, and interactions and to predict physicochemical properties responsible for bioactivity [95,96,97]. Owing to the parallel upgrades of computer hardware, more efficient procedures of conformational sampling, and more accurate calculations of potential energies, MD algorithms can now routinely simulate organic ligands, biomolecular receptors, and the entire cellular structures at the microsecond timescale [98,99,100,101,102]. All MD simulations calculate the potential energy *V* as a sum of bonded (stretching, bending, and torsion) and non-bonded (Coulomb and Lennard–Jones) interaction terms:(3)$$V_{bonded}=\underset{stretching}{\sum }K_{a}{\left(a-a_{o}\right)}^{2}+\underset{bending}{\sum }K_{\theta }{\left(\theta -\theta_{o}\right)}^{2}+\underset{torsion}{\sum }\underset{n}{\sum }K_{\phi }cos\left(n\phi -\gamma \right)$$
(4)$$V_{non-bonded}=\underset{Coulomb}{\sum }\frac{q_{i}\;q_{j}}{\varepsilon r_{ij}}+\underset{Lennard-Jones}{\sum }\left(\frac{A}{r_{ij}^{12}}-\frac{B}{r_{ij}^{6}}\right)$$
(5)$$V=V_{bonded}+V_{non-bonded}$$
where *a* is the bond length, *θ* is the bond angle, *a_o_* and *θ_o_* are their equilibrium values, respectively, *ϕ* is the torsion angle multiplied by an integer n, *γ* is the phase shift, *ε* is relative permittivity, *q_i_* and *q_j_* are electric charges, *r_ij_* are interatomic distances, and *K_a_*, *K_θ_*, *K_ϕ_*, *A*, and *B* are the constants.

These expressions constitute a molecular mechanics force field—a simplified, ball-and-spring approximation of quantum mechanical (QM) energies used in MD simulations for the evaluation of intra- and intermolecular interactions in large biomolecular systems. The straightforward assignment of these interactions is generally limited to the bond length and angle parameters, which can either be obtained by structure characterization or by QM geometry optimization. On the other hand, all the remaining interaction terms should be assigned separately for each pair of interacting atoms. Taking into account the empirical nature of the potential energy (Equations (3)–(5)), there can be no universally accepted force field, and the choice of which one is the most appropriate for the specific purpose is left to a simulation specialist. Among various types of force fields developed for MD simulations, the most frequently used for macrolide interaction studies are AMBER, CHARMM, and MMFF.

AMBER or ‘Assisted Model Building and Energy Refinement’ is a force field that was developed at Peter Kollman’s lab of the University of California, San Francisco in the 1980s [103]. Specific versions of the software are denoted AMBER ff YY-nn, where ‘ff’ stands for ‘force field’, ‘YY’ is the publication year, and ‘nn’ refers to an identifying nickname. In the AMBER parameterization strategy, small organic molecules, such as methanol, toluene, benzene, and *N*-methyl-acetamide serve as model compounds for calculating Lennard–Jones parameters [25]. Partial charges are calculated following a two-step procedure. Firstly, the electrostatic potential is minimized at the HF/6-31G* QM level of theory and approximated by least squares fitting. In the second step, equivalent charges are generated at freely rotating methyl groups, and the charges at non-hydrogen atoms are reduced until the modifications start to affect the overall fit. The described process is known as the restrained fitting of electrostatic potential [104]. Since the final model is constructed using non-iterative, modular parameterization, it can be adjusted by modifying the current steps or adding new into the established protocol. This is one of the features that has made AMBER an attractive force field for studying macrolides and their biomolecular receptors.

In their recent study [105], Gupta et al. utilized AMBER ff 99-SB to explain the regulatory activity of ErmBL mutants in the *E. coli* ribosomes bound to erythromycin or telithromycin. A combined use of MD simulations, toeprinting, and in vivo testing of the macrolide-specific induction of erythromycin-inducible ermC-β-galactosidase (pBLCLZα) reporters showed that the specificity of ribosomal response to chemical signals can be modulated by subtle changes in the nascent peptide. Hence, minor modifications in the amino acid sequence of ErmBL can either prevent stalling or induce the expression of antibiotic resistance genes in response to erythromycin, telithromycin, or both macrolides. These findings indicated that the sequences of many cellular proteins may have been evolutionarily optimized for making translation sensitive to environmental signals. Furthermore, Makarov et al. applied the same force field to investigate the transmission of allosteric signals between sensory elements in the exiting tunnel and the PTC [106]. A potential signal transduction pathway A2058–C2063 in the *E. coli* ribosome was explained by a dynamic ensemble of mutually dependent conformational states, where cascade-like changes can occur. Reversible inactivation of the PTC reflected structural rearrangement in the tunnel due to stacking contact between functionally important U2585 and C2063 residues.

Apart from ribosome studies, AMBER force fields have also been implemented in MD simulations of macrolide binding to regulatory proteins. Feng et al. studied erythromycin recognition by macrolide biosensor protein MphR(A), which triggers the expression of the *mphA* gene related to bacterial resistance [107]. Average contributions to the binding free energies provided evidence that Val66Leu/Val126Leu mutations increased the conformational flexibility of MphR(A) and reduced its binding affinity, resulting in lower resistance to erythromycin. It was supposed that the mutation might affect erythromycin binding to MphR(A) by changing the secondary protein structure.

Another force field package for MD simulations of macrolides is CHARMM or ‘Chemistry at HARvard Macromolecular Mechanics’. It was developed by Martin Karplus’s group of Harvard University in Cambridge, Massachusetts in the early 1980s [108]. Specific versions of this force field are named CHARMM-xx-Y-z, where ‘xx’ denotes the software edition, ‘Y’ specifies developmental, testing, or general release, and ‘z’ represents the minor revision number. In contrast to the fixed-step AMBER parameterization, CHARMM requires a multi-level, iterative optimization procedure [109]. Moreover, a complete scan of the potential energy surface (PES) should be carried out at a high QM level of theory. Lennard–Jones parameters can either be obtained from structural data and previous simulations or explicitly included in the optimization protocol. Additional calculations performed at potential hydrogen bond donors or acceptors with a single TIP3P solvent molecule allow the indirect encoding of molecular solvation properties into the force field parameters. Explicit modeling of the solvent-specific parameters is the major advantage of CHARMM in MD simulations of macrolides and biomolecules, since their behavior significantly depends on water-mediated interactions.

CHARMM parameters for erythromycin, azithromycin, and telithromycin can be determined using the force field toolkit (ffTK) in visual molecular dynamics (VMD), as reported by Pavlova and Gumbart [110]. The authors tested several methodologies for calculating partial atomic charges and fitting the dihedral angles. According to their simulations, the ffTK-derived partial atomic charges enabled the best improvement in conserving the interactions with ribosome. On the other hand, it was emphasized that the best fit of PES does not necessarily lead to the most accurate simulation results. The same parameters were implemented in MD simulations of several macrolide antibiotics and their derivatives in wild-type and modified *E. coli* ribosomes [111]. It was suggested that spatial displacement of the drug inside the exiting tunnel and its weaker interactions with modified A2058 nucleotide caused bacterial resistance in mutant, singly, and doubly methylated ribosomes. Further insight into the mechanisms of bacterial resistance induced by A2058G mutation and methylation at N6 of A2058 was obtained by combining grand canonical Monte Carlo (GCMC) and MD simulation methods [112]. Empirical calculations based on the CHARMM force field provided information on the interactions of telithromycin with the macrolide binding pocket and explained how the mutation of a single nucleotide affects the communication pathways relevant for ketolide activity.

MMFF or ‘Merck Molecular Force Field’ combines the best features of AMBER, CHARMM, and other force fields, making it equally suitable for small-molecule and macromolecular applications. The initial version of this force field, MMFF94 was published by Thomas A. Halgren at Merck Research Laboratories in Rahway, New Jersey in 1996 [113]. MMFF is focused on the systematic parameterization of conformational energies, intermolecular interactions, torsional barriers, and molecular geometries. Conformational energies and interaction parameters for MMFF are almost completely derived from computational data, due to a limited availability of high-quality structures [114]. On the other hand, vibrational frequencies are parameterized from a combination of theoretical and experimental data, but they are regarded less important. The MMFF parameterization strategy, which almost entirely relies on an accurate determination of mutually consistent parameters from computational data and ab initio calculations, has made this force field physically superior and applicable for interaction studies and computer-aided drug design [115].

A comprehensive study illustrating the potential of MMFF to explore the conformational behavior of rationally designed azithromycin and clarithromycin derivatives at physiological conditions was reported by Koštrun et al. [116]. Computational models based on the three-dimensional structure of macrolides were generated to explain structural, electronic, and conformational effects on measured properties and to predict the lipophilicity and cellular accumulation for this class of antibiotics from non-empirical parameters. MD simulations, descriptor calculations, and statistical analysis implied that the cellular accumulation was primarily affected by lipophilicity, basicity, and integy moment, which describes the distribution and separation of polar and non-polar regions. Different effects of substitution sites and substituent properties on a global physicochemical profile supported the fact that the macrolide structure is a complex unit and cannot be treated as a sum of smaller fragments, as their flexibility strongly depends on the macrocyclic ring and attached substituents.

Furthermore, MMFF can be implemented in conformational sampling methods, such as OMEGA. Poongavanam et al. applied OMEGA to search the conformational spaces of macrocyclic drugs and compared it with alternative MOELowModeMD (MOE) and MacroModel (MC) methodologies [117]. All three search engines generated solution-state conformers similar to those in the crystal structure, with OMEGA performing somewhat better in finding the conformation close to the crystal state, owing to the larger structural and property spaces sampled by this method. In addition, OMEGA was more consistent in predicting the effects of solvent polarity and behavior of conformationally flexible compounds. It was concluded that OMEGA should be considered complementary to MC and MOE, since it allows efficient conformational sampling and accurate prediction of macrolide properties relevant for their mechanism of action.

## 7. Conclusions

The advanced experimental and computational methods reviewed in this paper have successfully been applied to study the structure and interactions of macrolide antibiotics with their biological targets. It has been shown that the solution-state structures of macrolides, their receptors, and complexes are very similar to those observed in the crystalline state. However, the structural features may not exactly be the same in crystals and at or near physiological conditions, which has been elaborated. The presented structural models revealed that macrolide-ribosome interactions are species-specific, although the structural architecture of the complexes showed similarities. It is important to obtain structures with the highest resolution possible to elucidate all relevant structural details of the interactions. Another fact that should not be neglected is the conformational differences between macrolide complexes with the whole ribosome and with the large subunit alone, as observed for erythromycin by cryo-EM. Having this in mind, a complete insight into the mechanisms of complex formation can only be obtained by combining X-ray, cryo-EM, and NMR methods with MD simulations, biochemical and physicochemical assays. This is crucial for further research aiming at the design of new inhibitors and drugs.

## Figures and Tables

**Figure 1 ijms-21-07799-f001:**
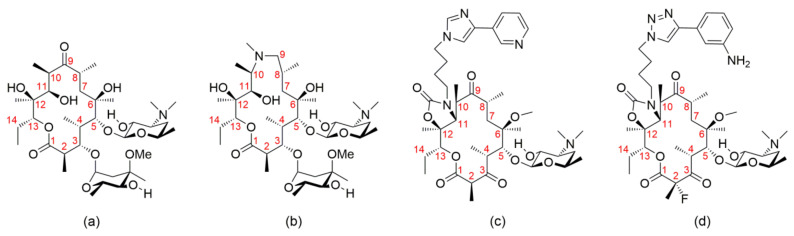
Representatives of four macrolide generations: (**a**) Erythromycin A; (**b**) Azithromycin; (**c**) Telithromycin; (**d**) Solithromycin with atom numbering of the macrolactone ring.

**Figure 2 ijms-21-07799-f002:**
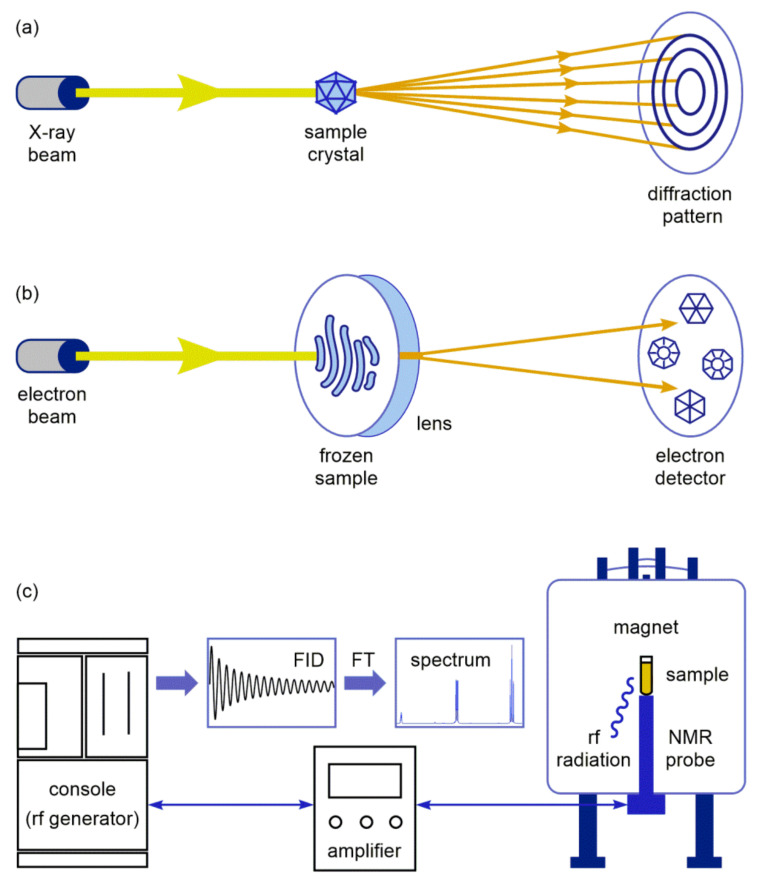
Basic principles of: (**a**) X-ray crystallography; (**b**) Cryogenic electron microscopy (Cryo-EM); (**c**) NMR spectroscopy.

**Figure 3 ijms-21-07799-f003:**
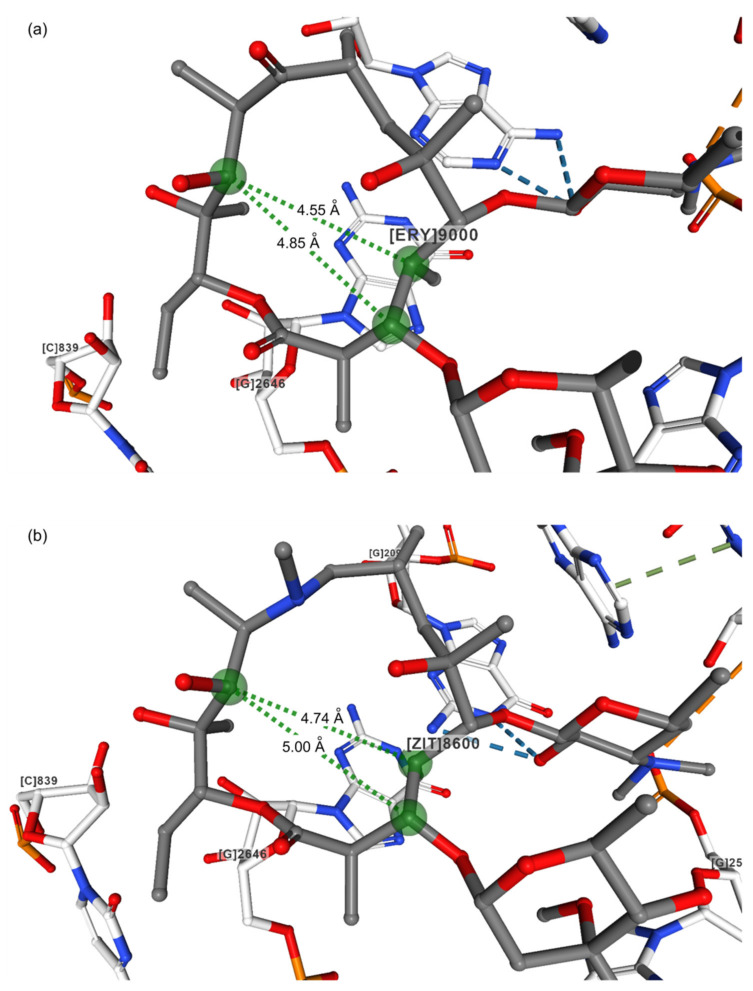
Crystal structures of (**a**) Erythromycin (PDB ID: 1YI2) [11,29]; (**b**) Azithromycin (PDB ID: 1M1K) [27,30] bound to the *H. marismortui* 50 S subunit.

**Figure 4 ijms-21-07799-f004:**
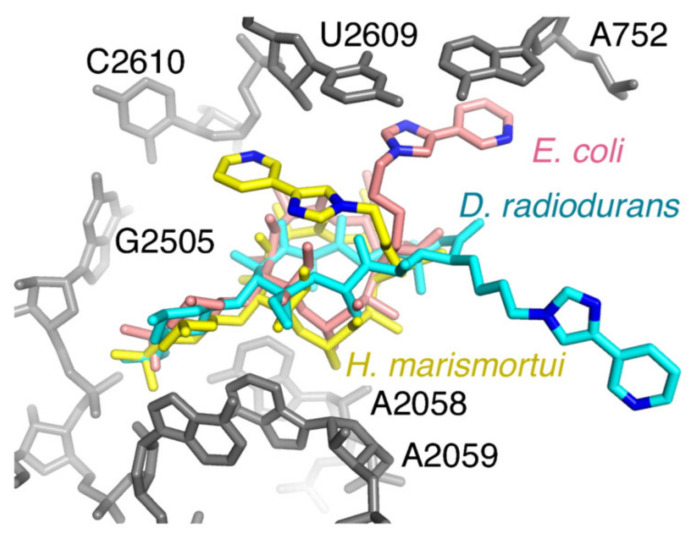
Telithromycin bound to the *E. coli* ribosome. A comparison of the conformations reported for telithromcyin bound to the ribosome. 23S rRNA for *E. coli* is shown in gray. Telithromycin models from *H. marismortui* (gold), *D. radiodurans* (cyan), and *E. coli* (pink) are shown. Nitrogens in the alkyl–aryl arm are also shown for reference; reproduced with permission by PNAS [12].

**Figure 5 ijms-21-07799-f005:**
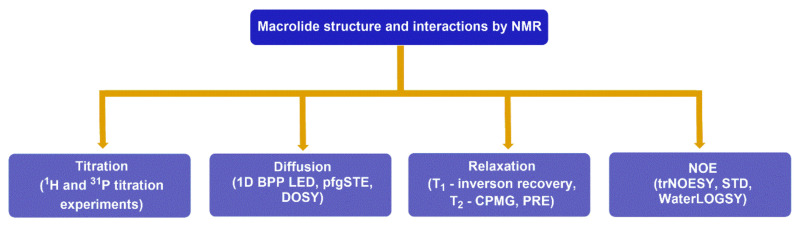
NMR experiments frequently used for studying the structure and interactions of macrolides.

**Figure 6 ijms-21-07799-f006:**
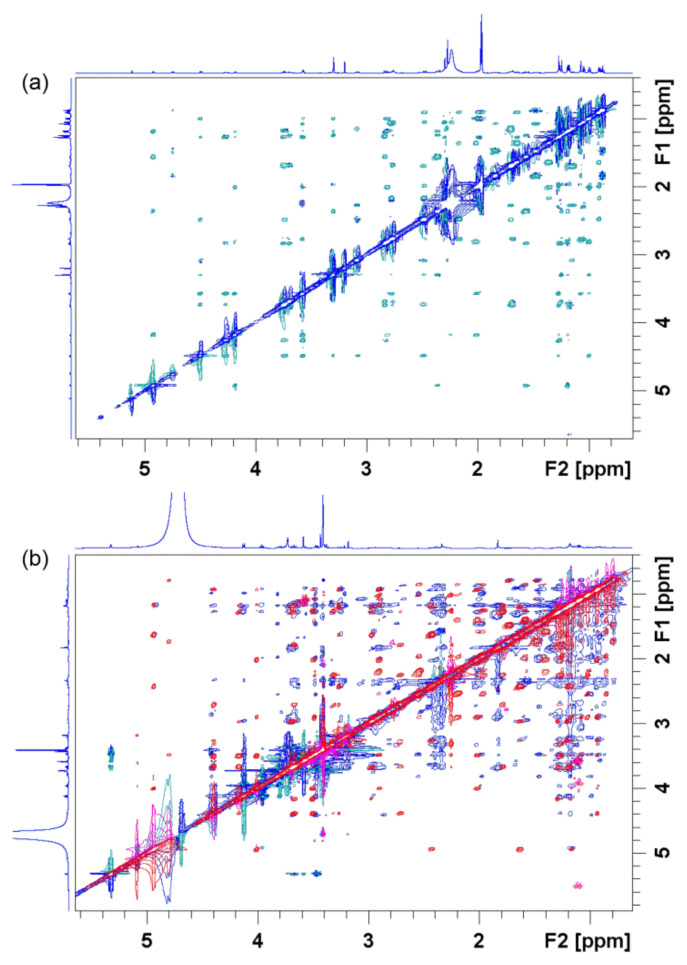
Nuclear Overhauser effect spectroscopy (NOESY) spectrum of 4″-aminopropyl azithromycin in (**a**) acetonitrile-d_3_ prior to addition of the *E. coli* ribosome compared with (**b**) NOESY (red and pink signals) and transferred nuclear Overhauser effect spectroscopy (trNOESY) spectrum (blue and green signals) of the same compound in Tris-d_11_ buffer (pH 7.4) prior to and after addition of the *E. coli* ribosome, respectively. The spectra were recorded at 25 °C and 600 MHz.

**Figure 7 ijms-21-07799-f007:**
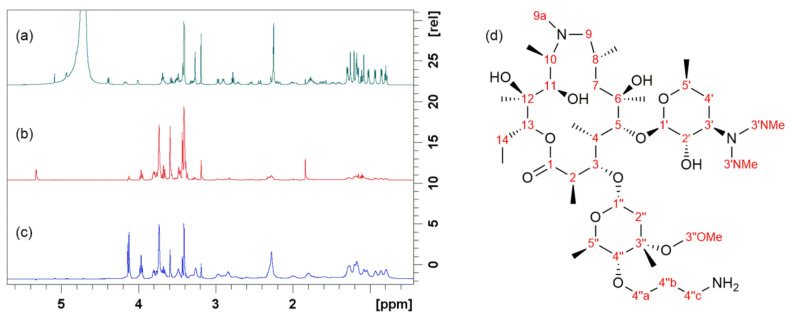
Binding of 4″-aminopropyl azithromycin to the *E. coli* ribosome studied by STD: (**a**) Proton spectrum prior to the ribosome addition; (**b**) Off-resonance spectrum after the ribosome addition; (**c**) Difference spectrum after the ribosome addition; (**d**) Chemical structure and atom numbering of 4″-aminopropyl azithromycin. The spectra were recorded at 25 °C and 600 MHz in Tris-d_11_ buffer (pH 7.4).

**Figure 8 ijms-21-07799-f008:**
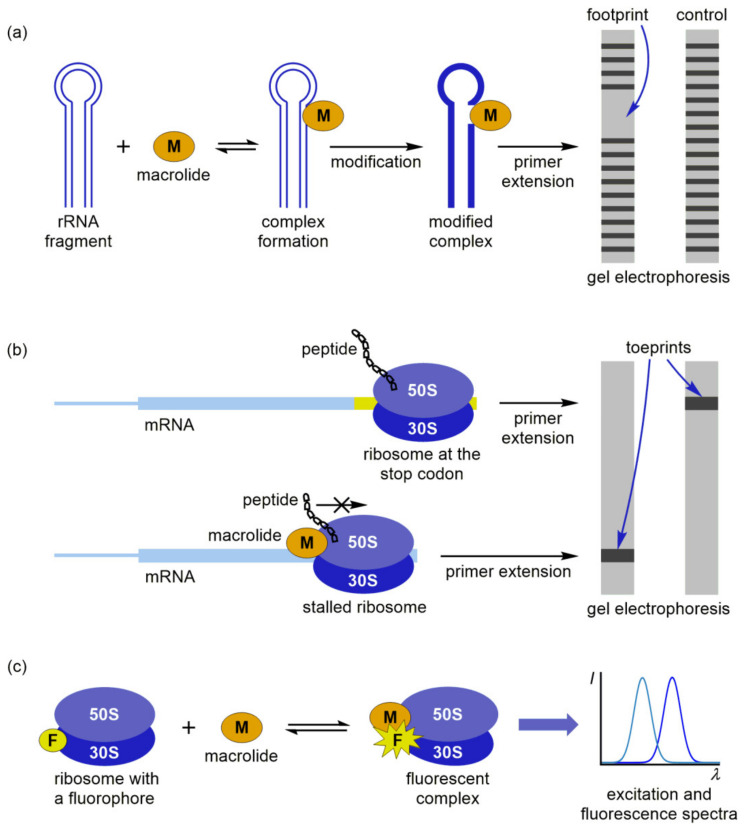
Macrolide binding to bacterial ribosome studied by: (**a**) Footprinting; (**b**) Toeprinting; (**c**) Fluorescence assays.

**Table 1 ijms-21-07799-t001:** Structural data for macrolides and their complexes with bacterial ribosomes in the crystal state.

Compound	Code	Resolution/Å	Distance/Å					
			C3–C11	C4–C11	C4–C6Me	C5–C6Me	C3–C8	C8–C11
Erythromycin, folded out ^1^	NAVTAF	0.85	4.69	4.33	3.90	2.52	5.50	3.65
Azithromycin, folded out ^1^	GEGJAD	0.82	4.86	4.60	3.92	2.50	5.74	4.80
Erythromycin bound to the *H. marismortui* 50S subunit [11] ^2^	1YI2	2.65	4.85	4.55	3.99	2.61	5.64	3.59
Azithromycin bound to the *H. marismortui* 50S subunit [11] ^2^	1YHQ	2.40	4.80	5.01	4.03	2.59	5.76	4.62
Azithromycin bound to the *H. marismortui* 50S subunit [27] ^2^	1M1K	3.20	5.00	4.74	3.92	2.55	5.82	4.84
Erythromycin bound to the *T. thermpohilus* ribosome [28] ^2^	4V7X	3.00	4.85	4.55	3.99	2.61	5.64	3.59
Azithromycin bound to the *T. thermpohilus* ribosome [28] ^2^	4V7Y	3.00	4.80	5.01	4.03	2.59	5.76	4.62

^1^ CCDC database. ^2^ PDB database.

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
