# Peer review of "Advanced Methods for Studying Structure and Interactions of Macrolide Antibiotics"

_ijms, 2020, doi:10.3390/ijms21207799_

Round 1

Reviewer 1 Report

The manuscript ijms-955404 is organized in a logical format. The number, quality and analysis of references are also adequate. Because of the differences between the styles of the different parts of the manuscript (see e.g. the different lengths of paragraphs) the readers have the feeling that the manuscript is not well written. I suggest unifying the style of the manuscript including the focus on the too short (one sentence) and too long (more than one page) paragraphs.

Author Response

We have followed the reviewer’s suggestion and unified the style of the manuscript, including the lengths of paragraphs.

Reviewer 2 Report

The manuscripts reviews an application of the experimental and computational tools to study interactions of the macrolide antibiotics with their biological targets. The advantages and drawbacks of x-ray crystallography, cryo-EM, NMR and computational methods are discussed in context of the macrolide antibiotics. The review largely represents a laundry list of structural method applied to the macrolide antibiotics interacting with different biological targets. There is no uniting theme or scientific conclusions made based on the analysis of the reviewed data.  The authors have not delivered “insight into the mechanisms of complex formation” declared in the conclusion.

Specific comments:

The authors incorrectly use “solid-state” terminology. Apparently, they mean crystal structures and perhaps cryo-EM structures. Crystals are not solid state – they contain high percentage of water and proteins in the crystals are hydrated. Even though mobility of protein is restricted in the crystal, protein is in a conformation sampled in solution. As the authors mentione in line 693, “the solution-state structures of macrolides, their receptors and complexes are very similar to those observed in the solid state. In cryo-EM, macromolecules are in conformations adopted in solution and selection of individual conformations occurs by image sorting using computer programs. Rephrase “solid-state” usage throughout the text.

In the discussion of the ribosome-bound macrolide conformations (see lines 126-141), the authors should discuss x-ray structures in context of the resolution. The resolution defines how credible are the boat/chair conformations assigned to the sugar moieties. PDB provides an opportunity to inspect electron density map to gain confidence that an assigned conformation is indeed obvious from electron density. In low-resolution structures, conformations of small molecules are often arbitrary chosen by the authors. It is also true for the conformations of the macrolactone ring and for the drug-target interactions (lines 142-160). The higher resolution, the more reliable is position of the Hetero-compound.

Line 178: high-resolution - what exactly is the resolution? Give numbers. It would give the reader perspective how reliable are the differences between the structures.

Author Response

  1. We have rephrased the term "solid-state" throughout the text
  2. We have changed the Conclusion part in order to further emphasize the importance of the presented techniques to elucidate ribosome-macrolide interactions
  3. We have added a paragraph on similarities and differences between free and bound macrolide conformations with respect to the resolution. Additional figure and table were inserted to explain details. We believe that this is a new contribution to the field. We completely agree with reviewer 2 that resolution is crucial to explain all the details of macrolide conformation and interactions and we stress this point also in conclusion.

Round 2

Reviewer 2 Report

In the revised manuscript, some previous criticism was addressed and an additional figure and a table were added to provide detains on the macrolide conformation in context of the crystal structure resolution.   

The authors should do a better job to replace the “solid-state” term throughout the text. Only in the abstract, it is used twice; also in lines 71, 97, 147, 191 and 322. It is a crystalline or crystal state, not a solid state.

Line 180: Poorly formulated sentence: When complexed with ribosome, the polar groups of the macrocyclic lactone ring are directed towards one side of the molecule while the other side is mostly hydrophobic.